# Spectral Analysis of Electricity Demand Using Hilbert–Huang Transform

**DOI:** 10.3390/s20102912

**Published:** 2020-05-21

**Authors:** Joaquin Luque, Davide Anguita, Francisco Pérez, Robert Denda

**Affiliations:** 1Dpto. Tecnología Electrónica, Universidad de Sevilla, Av. Reina Mercedes s/n, 41004 Sevilla, Spain; fperez@us.es; 2Department of Computer Science, Bioengineering, Robotics and Systems Engineering, University of Genoa, Via Opera Pia 13, I-16145 Genoa, Italy; davide.anguita@unige.it; 3Network Technology and Innovability, Enel Global Infrastructure and Networks, 00198 Rome, Italy; robert.denda@enel.com

**Keywords:** Hilbert–Huang Transform, Empirical Mode Decomposition, spectral analysis, electricity demand

## Abstract

The large amount of sensors in modern electrical networks poses a serious challenge in the data processing side. For many years, spectral analysis has been one of the most used approaches to extract physically meaningful information from a sea of data. Fourier Transform (FT) and Wavelet Transform (WT) are by far the most employed tools in this analysis. In this paper we explore the alternative use of Hilbert–Huang Transform (HHT) for electricity demand spectral representation. A sequence of hourly consumptions, spanning 40 months of electrical demand in Spain, has been used as dataset. First, by Empirical Mode Decomposition (EMD), the sequence has been time-represented as an ensemble of 13 Intrinsic Mode Functions (IMFs). Later on, by applying Hilbert Transform (HT) to every IMF, an HHT spectrum has been obtained. Results show smoother spectra with more defined shapes and an excellent frequency resolution. EMD also fosters a deeper analysis of abnormal electricity demand at different timescales. Additionally, EMD permits information compression, which becomes very significant for lossless sequence representation. A 35% reduction has been obtained for the electricity demand sequence. On the negative side, HHT demands more computer resources than conventional spectral analysis techniques.

## 1. Introduction

Modern electrical networks are one of the main scenarios in which a widespread deployment of sensors is being achieved [1], thus acquiring a large amount of information of different types. Processing that information fosters making more efficient and intelligent decisions, in such a way that resulting power networks are usually known as smart grids [2].

The large number of sensors used in today′s smart grids has become economically and technically feasible due to a significant reduction in their cost, the availability of higher capacity communication systems, and the greater accessibility to storage and processing equipment [3].

The straightforward availability of greater datasets poses a challenge on the data analytic side [4] making possible new and more efficient approaches to several applications such as fault detection [5], predictive maintenance [6], transient stability analysis [7], electric device state estimation [8], power quality monitoring [9], topology identification [10], renewable energy forecasting [11], and non-technical loss detection [12].

Especially relevant are the smart grid applications in which a high ratio of renewable and distributed energies has to be considered. Autoregressive integrated moving average (ARIMA) has been proposed in forecasting solar production [13], long-term energy demand [14] and electricity prices [15]. Other uses of data analytics in smart grids include heterogeneous data integration [16] or estimation of battery state-of-charge in solar farms [17]. A comprehensive and updated review of issues regarding the role of data analysis in the development of intelligent energy networks can be found in [18].

Among all smart grid applications, those concerning the load curve occupy a prominent place, for instance, load forecasting [19], load profiling [20], and load disaggregation [21,22]. Some of the most employed techniques for load curve analysis are frequency-domain based [23], in which mainly the Fourier Transform (FT) is applied to hourly [24], weekly [25], or annual [26] load curves. In those cases where short-term transitions have to be characterized, it is also common to rely on the Wavelet Transform (WT) for frequency analysis [27].

However, evaluating long sequences of energy demand poses a difficult challenge because, using classical spectral analysis, we can either focus on long-term behavior or, alternatively, pay attention to short-term evolutions. Coping with long sequences of data, we should be able to look at its overall evolution while simultaneously regarding seasonal variations and transient events. But in conventional spectral techniques, simultaneously managing different timescales is a task that remains unexplored and requires new solutions.

On the other hand, the use of Hilbert–Huang Transform (HHT) has gained increasing popularity in recent years for frequency analysis applied to many different realms, such as rolling bearing mechanical behavior [28], fault diagnosis of electrical motors [29], and even financial information [30]. HHT has also been applied to power networks in some specific tasks such as energy load forecasting [31] and fault classification [32]. However, to our best knowledge, the capabilities of HHT to discover and describe the ensemble of frequencies of a power network load profile remain unexplored.

The objective of this research is to obtain the multiple harmonic components of a long-term electricity load curve applying HHT, and to compare these results with those obtained using more traditional methods (FT and WT).

Spectral analysis techniques, especially HHT, play an important role in today’s practical problems of power networks with a high penetration of renewable energies. Recent studies have shown examples of statistical classical methods (like ARIMA) outperforming modern machine learning approaches [33], while other authors have reached the opposite conclusions [34]. But either using classical or modern forecasting methods, spectral decomposition of time series clearly provides better results [31,35].

This paper is organized in 5 sections. After this introduction, the first parts of Section 2 will present the dataset used to define the electricity load curve; then classical techniques for frequency analysis (FT and WT) will be reviewed. Section 2.3 will present newer approaches for spectral analysis based on Hilbert Transform (HT), while Section 2.4 will address the application of the Empirical Mode Decomposition (EMD) to solve the HT problems obtaining the so-called Hilbert–Huang Transform (HHT). In Section 3, results of applying classical and newer techniques to the dataset will be shown. Section 4 will discuss the principal findings of the research, comparing HHT with classical techniques in terms of frequency description, detection of abnormal behavior, and sequence information compression. Finally, Section 5 will provide some conclusions.

The main novel contributions which will be exposed throughout the paper address the problem of electricity demand multiscale spectral analysis:Revealing clearer, smoother, and more intuitive spectra, which discover more physically significant harmonics;enabling a more in-depth analysis of abnormal electricity demand at different timescales; andobtaining an improved lossless spectral compression method.

## 2. Materials and Methods

### 2.1. Dataset

Different national and international sources provide load curves at many levels of disaggregation. In this research, the Monthly Hourly Load Values (MHLV) for Spain have been used [36]. Maintained by the European Network of Transmission System Operators for Electricity (ENTSO-E) [37], the dataset contains hourly values of aggregated electricity demand for different European countries. By the time of its download, the dataset have values for the period comprised between the first of January 2016, and the end of April 2019, that is, 40 months (more than 3 years), with a total of N=29,183 values. Similar information can also be obtained from other sources [38].

The overall dataset is shown in Figure 1a, where the evolution of the demand along seasons can be easily seen. 

In Figure 1b, a ten week evolution (March and April 2016) is depicted, showing a typical weekly behavior with high electricity consumption on labor days and lower values on weekends. Finally, Figure 1c shows an example for a 15-day period, where the two daily peaks (typically at noon and mid-afternoon) are shown.

### 2.2. Conventional Techniques for Spectral Analysis

Frequency-domain analysis of the dataset is primarily tackled by employing Fourier Transform (FT). Calling x(t) the value of the electricity demand at time t, its frequency representation X(f) is defined as: (1)$$X\left(f\right)\equiv ℱT\left[x\left(t\right)\right]=\int_{-\infty }^{+\infty }x\left(t\right)\;e^{-j2\pi ft}\;dt.$$

When x(t) is digitized at $T_{s}$ intervals, that is, at a sampling rate $f_{s}=1/T_{s}$, a sequence of N values is obtained, where x(n) denotes the electricity demand at the n-th sampling time. For discrete sequences, frequency-domain representation is obtained using the Discrete Fourier Transform (DFT): (2)$$X\left(k\right)\equiv DℱT\left[x\left(n\right)\right]=\overset{N-1}{\underset{n=0}{\sum }}x\left(n\right)e^{-\frac{j2\pi }{N}kn},$$
where X(k) is a complex value representing the amplitude and phase of the harmonic component at frequency $k\;f_{s}/N$. The graph representing the absolute value |X(k)| is usually known as the spectrum of x. 

Instead of DFT, a faster version of the algorithm, called Fast Fourier Transform (FFT), is commonly used. As the dataset described in the previous subsection contains hourly load values, sampling period is $T_{s}=1\;h$ and, then, sampling frequency is $f_{s}=1\;{\mathrm{hour}}^{-1}=8760\;{\mathrm{year}}^{-1}$.

If x(n) cannot be considered a stationary sequence, it is more convenient to consider the DFT of a full sequence’s short slice: (3)X(m,k)≡STℱT[x(n)]=DℱT[x(n)·w(n−m)],
where w(n−m) is a certain window function centered at the m-th sampling time. In this research, the Hamming window function has been used [39]. X(m,k) is a discrete complex sequence that it is commonly represented by the square of its absolute value. The resulting graph is called the sequence’s spectrogram.

Although Short-Time Fourier Transform (STFT) is widely used in many technical fields, it suffers a serious limitation on time–frequency resolution. Calling $n_{w}$ the number of values (width) of the window function, then time interval Δt (time resolution) and frequency interval Δf (frequency resolution), are: (4)$$\Delta t=n_{w}T_{s},\;\;\;\;\Delta f=\frac{f_{s}}{n_{w}}.$$

Joint time–frequency resolution is limited because: (5)Δt·Δf=1,
that is, the greater the resolution in time, the smaller in frequency. Making window function wider (greater value of $n_{w}$), time interval Δt is increased (time resolution decreased) and frequency interval Δf is decreased (frequency resolution increased).

To overcome this problem many researchers use the Wavelet Transform (WT), defined as: (6)$$X\left(m,s\right)\equiv WT\left[x\left(n\right)\right]=\overset{N-1}{\underset{n=0}{\sum }}x\left(n\right)\psi \left[\frac{\left(n-m\right)T_{s}}{s}\right],$$
where ψ[·] is a time-limited function called a wavelet, which is shifted at time $mT_{s}$ (m is an integer) and time-scaled by a factor s (a real positive value). X(m,s) is a function, discrete in m and continuous in s, that it is commonly represented by the square of its value. The resulting graph is called the sequence’s scalogram where the m-axis represents the time and the s-axis is inversely proportional to the frequency. In this research, the Morlet wavelet has been used, which is defined as: (7)$$\psi \left[\xi \right]\equiv e^{-\frac{\xi^{2}}{2}}\cos\left(5\xi \right).$$

Wavelet transform is widely used for analyzing non-stationary sequences. In the case of stationary sequences, we are no longer interested in the evolution over time of its spectral behavior as they do not change. Thus, it is better to obtain a time-invariant spectrum using the Marginal Wavelet Transform (MWT) which is defined as: (8)$$X\left(s\right)\equiv ℳWT\left[x\left(n\right)\right]=\overset{N-1}{\underset{m=0}{\sum }}WT\left[x\left(n\right)\right]=\overset{N-1}{\underset{m=0}{\sum }}\overset{N-1}{\underset{n=0}{\sum }}x\left(n\right)\psi \left[\frac{\left(n-m\right)T_{s}}{s}\right].$$

### 2.3. The Hilbert Transform for Spectral Analysis

In some harmonic analysis applications, it is sometimes preferred to rely on an alternative approach called Hilbert Transform (HT), defined as: (9)$$\hat{x}\left(t\right)\equiv HT\left[x\left(t\right)\right]=\frac{1}{\pi }\int_{-\infty }^{\infty }\frac{x\left(\tau \right)}{t-\tau }d\tau \;,$$
where $\hat{x}\left(t\right)$ is a real-valued time-dependent function. When x is a discrete sequence x(n), Hilbert Transform can be written as:(10)$$\hat{x}\left(n\right)\equiv HT\left[x\left(n\right)\right]=\overset{N-1}{\underset{\begin{array}{c} m=0\\ m\neq n \end{array}}{\sum }}\frac{x\left(m\right)}{n-m}.$$

In most applications, the resulting Hilbert Transform $\hat{x}\left(n\right)$ is combined with the original function, obtaining the so-called analytic function, defined as:(11)$$\tilde{x}\left(n\right)\equiv x\left(n\right)+j\;\hat{x}\left(n\right),$$
which is a time-dependent complex sequence. Its module is called instantaneous amplitude, that is, $𝕒\left(n\right)\equiv \left|\tilde{x}\left(n\right)\right|$. An instantaneous frequency 𝕗(n) can also be obtained using the phase of the analytic function: (12)$$𝕗\left(n\right)\equiv \frac{\Delta \arg\left[\tilde{x}\left(n\right)\right]}{T_{s}}=\frac{\arg\left[\tilde{x}\left(n\right)\right]-\arg\left[\tilde{x}\left(n-1\right)\right]}{T_{s}}.$$

Combining the instantaneous amplitude and frequency in a single vector of two functions, the Hilbert Spectrum (HS) is obtained, that is: (13)S(n)=HS[x(n)]≡[𝕒(n),𝕗(n)]∈ℝ2.

For the case of stationary sequences, it is better to obtain the Marginal Hilbert Spectrum (MHS), which is defined as: (14)$$S_{m}\left(f\right)\equiv \overset{N-1}{\underset{\begin{array}{c} n=0\\ 𝕗\left(n\right)=f \end{array}}{\sum }}𝕒\left(n\right).$$

### 2.4. Empirical Decomposition and the Hilbert–Huang Transform for Spectral Analysis

In many practical sequences it is not uncommon that some Hilbert instantaneous frequencies (and their derived HS and MHS) have negative values, which are meaningless [40]. For the instantaneous frequency to always be positive, the sequence must satisfy two conditions [41]:It is symmetric with respect to a null local average value; andIt has the same number of zero-crossing and extrema.

A sequence fulfilling these requirements is called an Intrinsic Mode Function (IMF). Therefore, to obtain meaningful Hilbert Transform, the original sequence x(n) should be first decomposed in several IMFs $c_{i}\left(n\right)$. A method to achieve this goal is named Empirical Mode Decomposition (EMD) which is based on two nested loops: An outer iteration to obtain the i-th IMF $c_{i}$(n); and an inner iteration to obtain $h_{i,j}\left(n\right)$, the j-th trial for the i-th IMF. The EMD algorithm (also called a sifting process) can be summarized in the following steps [42]:Consider original data as the first residue: $r_{1}\left(n\right)=x\left(n\right)$Make i=1Obtaining the i-th IMF: $c_{i}\left(n\right)$
Consider the i-th residue as the first trial for the i-th proto-IMF: $h_{i,1}\left(n\right)=r_{i}\left(n\right)$Make j=1Consider the j-th trial for the i-th proto-IMF as the ongoing sequence: $x_{i,j}\left(n\right)=h_{i,j}\left(n\right)$Obtain upper local extrema of $x_{i,j}\left(n\right)$Obtain upper envelope (joining upper extrema with a cubic spline): $u_{i,j}\left(n\right)$Obtain lower local extrema and lower envelope: $l_{i,j}\left(n\right)$Obtain mean value of upper and lower envelopes: $m_{i,j}\left(n\right)=\left[u_{i,j}\left(n\right)+l_{i,j}\left(n\right)\right]/2$Obtain the j-th trial for the i-th proto-IMF subtracting the mean value from the original data: $h_{i,j}\left(n\right)=x_{i,j}\left(n\right)-m_{i,j}\left(n\right)$Repeat inner steps 3 to 7 (increasing j) until the inner loop stop criterion is fulfilled, which occurs in the k-th inner iterationConsider the last proto-IMF $h_{i,k}\left(n\right)$ as the i-th IMF: $c_{i}\left(n\right)=h_{i,k}\left(n\right)$Inner loop stop criterion: During 𝑆 consecutives iterations the number of upper extrema ($U_{e}$), lower extrema $\left(L_{e}\right),$ and zero-crossing $(Z_{c}$) of the ongoing sequence $x_{i,j}\left(n\right)$ satisfy the equation $|U_{e}+L_{e}-Z_{e}|\leq 1$
, where 𝑆 is a predefined constant (usually 4≤S≤8)Obtain the i-th residue subtracting i-th IMF from the first trial of the ongoing sequence: $r_{i}\left(n\right)=x_{i,1}\left(n\right)-c_{i}\left(n\right)$Repeat outer steps 3 and 4 (increasing i) until the outer loop stop criterion is fulfilled, which occurs in the q-th outer iterationOuter loop stop criterion: Residue is a monotonic function; or IMF or residue are smaller than a predetermined valueConsider the last residue as the overall residue: $r\left(n\right)=r_{q}\left(n\right)$.

Finally, using EMD, the sequence x(n) can be decomposed into a sum of q IMFs and a residue, according to the following expression:(15)$$x\left(n\right)=\left(\overset{q}{\underset{i=1}{\sum }}c_{i}\left(n\right)\right)+r\left(n\right).$$

Considering the way of obtaining the IMFs, they satisfy the conditions so that their HSs do not have meaningless negative instantaneous frequencies. Calling ${𝕒}_{i}\left(n\right)$ and ${𝕗}_{i}\left(n\right)$ the instantaneous amplitude and frequency of the i-th IMF, the Hilbert–Huang Transform (HHT) is defined as the ensemble of the Hilbert Spectrum (HS) of each IMF and the residue, that is, q+1 vectors each of them of two functions:(16)$$HHT\left[x\left(n\right)\right]\equiv \left\{HS\left[c_{1}\left(n\right)\right],HS\left[c_{2}\left(n\right)\right],\cdots ,HS\left[c_{q}\left(n\right)\right],HS\left[r\left(n\right)\right]\right\}.$$
(17)$$HHT\left[x\left(n\right)\right]=\left\{\left[{𝕒}_{1}\left(n\right),{𝕗}_{1}\left(n\right)\right],\left[{𝕒}_{2}\left(n\right),{𝕗}_{2}\left(n\right)\right],\cdots ,\left[{𝕒}_{q}\left(n\right),{𝕗}_{q}\left(n\right)\right],\left[{𝕒}_{q+1}\left(n\right),{𝕗}_{q+1}\left(n\right)\right]\right\}.$$

In these equations the residue is understood as the (q+1)-th IMF. This function, which is discrete in time but continuous in frequency and amplitude, is also known as the Hilbert–Huang Spectrum (HHS).

To manage HHS as an ensemble of q+1 pairs of vectors is often quite burdensome and leads to graphics that are not easy to read. To simplify its representation it is common to discretize the frequencies using a quantizing Δf step, building up a grid in the time–frequency plane. Then the amplitudes of all IMFs for every single grid spot are added, obtaining a Discrete Hilbert–Huang Spectrum (DHHS) defined by:(18)$$X\left(n,k\right)\equiv DHHS\left[x\left(n\right)\right]=\overset{q+1}{\underset{\begin{array}{c} i=1\\ k\Delta f\leq {𝕗}_{i}\left(n\right)<\left(k+1\right)\Delta f \end{array}}{\sum }}{𝕒}_{i}\left(n\right).$$

For the case of stationary sequences, it is sometimes better to obtain the Marginal Hilbert–Huang Spectrum (MHHS), which is derived from DHHS and defined as:(19)$$S_{m}\left(k\right)\equiv \overset{N-1}{\underset{n=0}{\sum }}X\left(n,k\right)=\overset{N-1}{\underset{n=0}{\sum }}\overset{q+1}{\underset{\begin{array}{c} i=1\\ k\Delta f\leq {𝕗}_{i}\left(n\right)<\left(k+1\right)\Delta f \end{array}}{\sum }}{𝕒}_{i}\left(n\right).$$

## 3. Results

### 3.1. Spectral Analysis of Electricity Demand

In this subsection the spectral analysis methods previously explained will be applied to the MHLV load curve for Spain.

#### 3.1.1. Fourier Transform of Electricity Demand

Let us begin applying FT to the dataset, resulting in the spectrum depicted in Figure 2a. As the sampling frequency is $f_{s}=8760\;{\mathrm{year}}^{-1}$, full spectrum contains meaningful frequencies up to $f_{s}/2=4380\;{\mathrm{year}}^{-1}$. Regardless of the Direct Current (DC) component (corresponding to the energy demand mean value), two main harmonics can be seen: At 365 (times per year) corresponding to once per day; and at 730 corresponding to twice per day. These two harmonics reveal the main periodic behavior of the load curve featured by a daily evolution with two humps at noon and mid-afternoon. 

Zooming in on the spectrum to discard higher frequencies (Figure 2b), a third harmonic stands out at the frequency of 52 (times per year), revealing a weekly periodicity characterized by a higher demand on labor days and lower consumption on weekends. A much closer zoom centered on the low frequency spectrum (Figure 2c) shows a harmonic peak at frequency 2 (twice per year) and a lower and not very well defined peak at 1 (once per year), corresponding to a yearly periodicity with two seasonal peaks of high demand (due to heating in winter and air conditioning in summer). It has to be noted that this graph has a frequency resolution which can be computed by:(20)$$\Delta f=\frac{f_{s}}{N}=\frac{8760}{29183}=0.3\;years^{-1}.$$

#### 3.1.2. Short-Time Fourier Transform of Electricity Demand

Trying to discover transient behaviors, an STFT with time resolution Δt=1 week has been applied to the dataset’s sequence, obtaining the result depicted in Figure 3a. It can be seen that the spectrogram is quite stationary, that is, it has approximately the same values along time, although a small decrease during summer can be noted (mainly in higher frequencies). The most significant harmonics (dark red in the graph) appear in the low-frequency band, and a zoom-in of the frequency axis is shown in Figure 3b, where several harmonics clearly appear (around 50, 400, and 750 $\mathrm{years}{}^{-1}$). However, their values are not precisely determined as the corresponding frequency resolution is, according to Equation (5), $\Delta f=52\;\mathrm{years}{}^{-1}$. 

To increase frequency resolution, the width of the window function has to be enlarged and, consequently, the time resolution decreases. Figure 3c depicts the result when a Δt=0.5 years is applied, resulting in a frequency resolution of $\Delta f=2\;\mathrm{years}{}^{-1}$.

#### 3.1.3. Wavelet Transform of Electricity Demand

To overcome the limitations of STFT regarding the time–frequency resolution, a Morlet WT is applied to the sequence according to Equation (6), obtaining the scalogram shown in Figure 4a. For an easier reading, the vertical axis has been converted from scale s to frequency. Again, the electricity demand frequency representation shows a quite stationary behavior, although a small decrease during summer can be noted in the highest frequencies. Several harmonics clearly appear in the plot, centered around 50, 400, and 750 $\mathrm{years}{}^{-1}$. Figure 4b presents a detail of the scalogram for the lower frequencies.

#### 3.1.4. Marginal Wavelet Transform of Electricity Demand

If the sequence analysis considers only its stationary behavior, it is better to obtain the marginal spectrum applying Equation (8). The result is shown in Figure 5 using different graphic scales and with different frequency resolution. The accumulated over time spectral amplitude (vertical axis) versus frequency (horizontal axis) is plotted. Depicted in red are those harmonics already identified in Figure 2, while the values in green reveal new harmonics.

#### 3.1.5. Hilbert Spectrum of Electricity Demand

Now that the more conventional spectral analysis techniques have been employed, let us focus on the application of Hilbert Transform to our dataset. In Figure 6a, Hilbert Spectrum is depicted in accordance with the definitions in Equation (13). 

The plot represents the instantaneous frequency (vertical axis) versus time (horizontal axis), while the instantaneous amplitude corresponding to each time is also color-coded. As there is an instantaneous amplitude-frequency pair at every hour, the total number of points (pairs) in the full-scale graph is quite high (N=29,183). Therefore, the figure is easier to understand if the pairs are drawn as scattered points, that is, not forming a line.

On the other hand, Figure 6b shows the Marginal Hilbert Spectrum as it is defined in Equation (14). The accumulated-over-time amplitude (vertical axis) versus instantaneous frequency (horizontal axis) is plotted. In both figures, meaningless negative frequencies clearly appear, making a physical interpretation of these spectra difficult. 

#### 3.1.6. Hilbert–Huang Spectrum of Electricity Demand

To tackle the Hilbert Spectrum’s lack of meaning, the original electricity demand sequence is split into several Intrinsic Mode Functions (IMF) using the Empirical Mode Decomposition (EMD) method described in Section 2.4. The result of this process is a set of 12 IMFs and a residue, which are depicted in Figure 7 using a monthly timescale. The corresponding original signal is shown in the first panel and in Figure 1a. The last IMFs (those with higher indexes) contain the electricity demand slow-rate evolution. Therefore, IMFs 12, 11, 10, and 9 show an approximate period of 30, 12, 6, and 2 months, respectively. It can be seen that, for instance, the higher demands in winter and summer are neatly uncovered by the peaks in IMF 10 (6-month period).

For a better understanding of the IMFs behavior, they are also represented in a weekly timescale. Figure 8 depicts a 10-week sequence corresponding to the time-domain representation of the original signal shown in the first panel and in Figure 1b. It shows that IMFs 8, 7, 6, and 5 have approximate periods of 4, 2, 1, and 0.5 weeks, respectively. It is not difficult to establish connections between the original signal and IMFs. For instance, IMF 5 (0.5-week period) clearly presents two singularities (very low values) at weeks 12 and 17, corresponding to two quite visible depressions of energy demand at these times in the original signal. Additionally, the deepest depression in the original signal during this 10-week period occurs at week 12, which is also clearly revealed by the minimum value of IMF 8 (4-week period).

Finally, Figure 9 depicts a 15-day sequence corresponding to the time-domain representation of the original signal shown in the first panel and in Figure 1c. It reveals approximate periods of 4, 2, 1, and 0.5 days for IMFs 4, 3, 2, and 1, respectively. Again, connections between original signal and IMFs can be easily established. For instance, IMF 1 (0.5-day period) presents very low amplitude corresponding to two low energy demand periods during weekend days 16 and 23, both clearly visible in Figure 1b. These weekend days can also be detected by low amplitude in IMF 2 (1-day period).

After the EMD process, every resulting IMF does satisfy the conditions expressed in Section 2.4 for having a meaningful Hilbert Transform (HT). The electricity demand Hilbert–Huang Transform (HHT) is then no more than the ensemble of the HT of each IMF, as reflected in Equation (17), which is depicted in Figure 10a. The plot represents, for each IMF, the instantaneous frequency (vertical axis) versus time (horizontal axis), while the instantaneous amplitude corresponding to each point is also color-coded. This plot can be seen as 13 overlapping Hilbert Spectra (corresponding to each of the 12 IMFs plus the residue). As the graph covers the full timescale, it is drawn as a scatter plot for a better interpretation.

On the other hand, Figure 10b shows Marginal Hilbert–Huang Spectrum as it is defined in Equation (19). The accumulated-over-time amplitude (vertical axis) versus instantaneous frequency (horizontal axis) is plotted, with all the IMFs added.

In both figures, meaningless negative frequencies have almost disappeared, presenting only some negligible values which will be discarded from now on. 

When the timescale is smaller, the points in the Hilbert–Huang Spectrum (HHS) can be joined in a line plot without affecting its readability. That is the case in Figure 11a where 13 lines are drawn, one per IMF. Each line represents the instantaneous frequency versus time for a single IMF, with color representing the instantaneous amplitude. It can be seen that the most significant components are in the low frequency range with two outstanding regions, around twice (IMF 1) and once (IMF 2) per day (365 and 730 times per year). It can also be noted that higher frequencies (but with low amplitudes) exist on IMF 1 during weekends. 

Reading an HHS, either in its scatter or set-of-lines plot versions, is sometimes difficult due to the fact that several IMFs overlap in the graphic representation (as in Figure 10a and Figure 11a). Therefore, it is a common practice to discretize frequencies to obtain the Discrete Hilbert–Huang Spectrum (DHHS) as it is defined in Equation (18). The result is depicted in Figure 11b where again the once and twice per day frequencies can be seen.

The MHHS can now easily be extended to the whole dataset obtaining the results depicted in Figure 12a, where the red arrows on the right axis point to a stationary and well-defined 365 times per year frequency (once per day), as well as a somehow fluctuant 730 frequency (twice per day). As most of the harmonics are in the low frequency region, Figure 12 (panels b to f) shows the MHHS at different frequency (vertical) scales. The most outstanding regions are pointed with an arrow at the following panels and frequencies: (b) 365; (c) 52 (once per week); (d) 2 (once per semester), and a quite fluctuant component between 4 (once per trimester) and 6 (once per bimester); (e) 1 (once per year) and 2 (once per semester); and (f) 0.3 (once per full 40 month period) and 1 (once per year).

If the electricity demand can be considered stationary, or if we are not interested in its transient behavior, the temporal axis of DHHS can be squeezed to obtain the Marginal Hilbert–Huang Spectrum (MHHS) according to Equation (19). Results obtained using different frequency scales are depicted in Figure 13. Besides the harmonics obtained by FFT (marked in red), the Hilbert–Huang analysis reveals new harmonics pointed in green in the graphics. The new values, in times per year, are the following: 183 (2-day period); 91 (4-day period); 12 (1-month period); 4.35 (12-week period) and a blurry region till 6 (2-month period); and 0.3 covering the full 40-month dataset. 

It has to be noted that although DHHS and its derived MHHS are quantized in frequency (also in time), the frequency step Δf can be arbitrary selected. To maintain the details at each frequency scale, the number of frequency quanta has been fixed to 100 in every panel of Figure 13 (also in Figure 12). On the other hand, frequency resolution in FT is fixed and depends on the number of values in the dataset (Δf=0.3 in our case). Then FT leads to an over-detailed spectra for full range frequencies (Figure 2a), and to an under-detailed spectra for small range frequencies (Figure 2c). However, HHT produces spectra with a selectable frequency granularity which can be adjusted to obtain similar detail levels at each frequency range, as can be seen in Figure 13.

### 3.2. Spectral Analysis of the Intrinsic Mode Functions

One of the key elements of Hilbert–Huang Spectrum is the decomposition of the original sequence (the electricity demand in our case) in an ensemble of Intrinsic Mode Functions (IMFs). Although the main goal of this decomposition is to obtain a meaningful Hilbert Spectrum, the IMFs have interest on their own, as any of them convey information about the electricity demand evolution for a certain periodicity (or frequency) scale. 

For this reason, it is worthy to study the spectral representation of every individual IMF. The first approach will be to apply Fourier Transform (FT), getting the results depicted in Figure 14, where the spectrum for the original sequence, the 13 IMFs, and the residue, can be compared.

In each panel, the red circle marks the peak frequency, and the dashed green line represents the spectrum centroid defined as: (21)$$f_{c}\equiv \frac{\sum_{k=1}^{N}kf_{s}X\left(k\right)}{\sum_{k=1}^{N}X\left(k\right)},$$
where $\;f_{s}$ is the sampling ratio and X(k) is the DFT of the IMF obtained applying Equation (2). It can be seen that centroid is in all cases at a higher frequency than the peak, which indicates that the spectrum contains a significant right-side tail.

Our second approach for the spectral characterization of IMFs will be to apply Hilbert Transform recalling that, by definition, IMFs verify the conditions to avoid meaningless negative frequencies. Then, the Marginal Hilbert Spectrum (MHS) is obtained according to Equation (14) and the results are shown in Figure 15. 

Unlike DFT, peaks and centroids have similar values in MHS, indicating that spectra are approximately balanced around their central values. Additionally, it can be noted that frequency peaks are quite similar in both methods.

Spectra for every IMF can also be plotted as a color map with frequency (vertical axis) versus IMF index (horizontal axis) color-coding the spectrum value. Figure 16a shows the results applying DFT, while MHT results are depicted in Figure 16b. Peaks can be identified as the highest colored regions for each IMF, while the centroid is plotted as a black line.

In the effort to obtain the frequency (or period) that best defines every IMF, the use of the autocorrelation will finally be explored, defined as: (22)$$\varrho \left(L\right)=\frac{\mathrm{cov}\left[x\left(n\right),x\left(n-L\right)\right]}{\mathrm{var}\left[x\left(n\right)\right]}=\frac{\sum_{n=1}^{N}\left[x\left(n\right)-\mu_{x}\right]\left[x\left(n-L\right)-\mu_{x}\right]}{\sum_{n=1}^{N}{\left[x\left(n\right)-\mu_{x}\right]}^{2}},$$
where x(n) denotes the electricity demand series, $\mu_{x}$ is its mean value, and ϱ(L) is the Pearson autocorrelation coefficient for a certain lag L (an integer number). Then, the main periodicity is defined by $L_{p}\Delta t$, where $L_{p}$ is the lag corresponding to the autocorrelation first peak. Its inverse can be considered the dominant frequency according to:(23)$$f_{d}\equiv \frac{1}{L_{p}\Delta t}.$$

The autocorrelation corresponding to the first IMF is shown in Figure 17a, where the resulting 12 h main periodicity is marked with a red circle.

In this subsection, three methods to obtain the featuring frequency of every IMF have been presented. The results are summarized and compared in Figure 17b. It can be seen that all of them offer very similar results, except the DFT-obtained centroids. A more detailed analysis shows that the IMF main frequency is approximately halved in every step, as is clearly indicated by the dashed red line in the plot.

### 3.3. Statistical Analysis of the Intrinsic Mode Functions

To get a deeper insight on IMFs, a statistical analysis of its Hilbert Transforms will now be developed. Let us consider the electricity demand sequence x(n) and its Empirical Mode Decomposition (EMD). Let us call $z_{i}\left(n\right)$ the resulting i-th IMF and let us obtain its Hilbert Spectrum $S_{i}\left(n\right)=HS\left[z_{i}\left(n\right)\right]\equiv \left[{𝕒}_{i}\left(n\right),{𝕗}_{i}\left(n\right)\right]$ according to Equation (13). If the instantaneous frequency ${𝕗}_{i}\left(n\right)$ is considered a random variable, its probability distribution function can be derived as:(24)$$F_{{𝕗}_{i}}\left(\varphi_{i}\right)\equiv \mathrm{Prob}\left[{𝕗}_{i}<\varphi_{i}\right]=\frac{1}{N}\overset{N}{\underset{\begin{array}{c} i=1\\ {𝕗}_{i}<\varphi_{i} \end{array}}{\sum }}1.$$

Then, its derivative is called the Probability Density Function (PDF), commonly approximated by a histogram, and defined as:(25)$$f_{{𝕗}_{i}}\left(\varphi_{i}\right)\equiv \frac{dF_{{𝕗}_{i}}\left(\varphi_{i}\right)}{d\varphi_{i}}=\mathrm{Prob}\left[\varphi_{i}\leq {𝕗}_{i}<\varphi_{i}+d\varphi_{i}\right]=\underset{\Delta \varphi_{i}\to 0}{\lim}\frac{1}{N\Delta \varphi_{i}}\overset{N}{\underset{\begin{array}{c} i=1\\ \varphi_{i}\leq {𝕗}_{i}<\varphi_{i}+\Delta \varphi_{i} \end{array}}{\sum }}1.$$

The resulting PDF for the instantaneous frequencies of every IMF is depicted in Figure 18. In each panel, the red circle and the dashed green line marks mode and mean, respective of the instantaneous frequency. These results are similar to those obtained through the spectral analysis shown in Figure 15.

Besides instantaneous frequency, the instantaneous amplitude ${𝕒}_{i}\left(n\right)$ can also be analyzed as a random variable, and its probability density function can be derived in a similar way. The resulting amplitude PDFs $f_{{𝕒}_{i}}$ (vertical axis) versus instantaneous amplitude (horizontal axis) are depicted in Figure 19 (blue line). For the sake of completeness, the same figure also depicts in a rotated way (orange line), the frequency PDFs $f_{{𝕗}_{i}}$ (horizontal axis) versus instantaneous frequency (vertical axis).

Probability Density Functions for every IMF can also be plotted as a color map with instantaneous amplitude (vertical axis) versus IMF index (horizontal axis), color-coding the PDF value (normalized in the 0–1 range), as is shown in Figure 20a. A similar plot for instantaneous frequency PDF is depicted in Figure 20b. Peaks (mode) can be identified as the highest colored regions for each IMF, while mean value is plotted as a black line.

Previous statistical analyses consider instantaneous amplitude and frequency as two independent variables, but this is not really the case. Extending the PDF concept to the full Hilbert Spectrum considered as a 2-dimensional random process  S(n)=[𝕒(n),𝕗(n)], a joint PDF can be obtained, and the results for each IMF are shown in Figure 21. They are plotted color-coding the logarithm of the joint PDF (normalized in the 0–1 range) as a function of the instantaneous amplitude (horizontal axis) and frequency (vertical axis). The mean value $\left[\mu_{𝕒},\mu_{𝕗}\right]$ is marked with a black cross.

It can be seen that some IMFs present a very well defined frequency, for example, IMF 3 at 365 times per year, IMF 6 at 52, or IMF 11 at once per year. Some others, for instance IMF 2 and 5, show this defined frequency at 365 and 52, plus a secondary lower-amplitude higher frequency region at around 730 and 91, respectively.

## 4. Discussion

This section will provide an interpretation of the previous results and a comparison of HHT with more conventional spectral analysis techniques, such as Fourier or Wavelet Transforms. We will focus on three aspects: The ability of HHT to provide more physically accurate frequency detection; its application to abnormal behavior; and its power as an information compression technique.

### 4.1. Reading HHT Spectrum

Comparing spectral analysis by using HHT, as is depicted in Figure 13, and results obtained using conventional FFT, as in Figure 2, clearly shows that HHT provides more clear and meaningful spectra. HHT is also capable of discovering new underlying frequencies, such as 91 and 183 times per year. 

Moreover, FFT-based spectrum frequency resolution is limited by the sampling frequency and the number of values, as it is described in Equation (20), while HHT-based spectra can arbitrarily choose frequency resolution by freely defining the quantizing Δf step in Equation (18). This HHT feature is especially powerful to detect low frequency components (long-term periods) of electricity demand.

Wavelet Transform (WT), as it is shown in Figure 5, also enhances FFT low-frequency resolution. However, compared to HHT, WT is not as powerful at discovering new frequencies, and they are less accurate [43].

Therefore, HHT-based spectral analysis results are clearer, more intuitive, and more physically meaningful than FFT and even WT. Other authors have also obtained similar results studying electrical users [44], and other electrical or mechanical sequences [45,46].

### 4.2. Detecting Abnormal Energy Demand Behavior

Although full research on abnormal detection is out of the scope of the present paper, some of the findings in our research also have significant implications in detecting electricity demand abnormalities. 

The full sequence decomposition using EMD provides an ensemble of IMFs of different periodicities (or frequencies). Therefore, by choosing the proper IMF, the abnormal behavior at the IMF periodicity scale can be detected. For instance, IMF 6 has a periodicity of about one week (see Figure 15); thus, examining this IMF, it would be possible to detect weeks with abnormal electricity demand. Similarly, as IMF 3 has a periodicity of one day, abnormal daily consumptions would be detected based on its analysis. EMD thus provides a tool for searching for abnormalities at a determined timescale.

On the other hand, the Hilbert Spectrum of each IMF provides instantaneous amplitude and frequency which can be labeled as normal or abnormal using different techniques. As a first approach, amplitude (or frequency or joint) Probability Density Function (PDF) can be used as the likelihood of normal behavior. Detecting low likelihood is therefore equivalent to detecting abnormal demand. Figure 22 shows the instantaneous amplitude logarithmic likelihood for every IMF. 

Considering, for instance, IMF 6, a very low likelihood can be appreciated at approximately time 2.0 (years), corresponding to New Year’s week. As this IMF has a weekly periodicity, it should correspond to a week with abnormal electricity demand. Let us take a closer look, just as an example, at this IMF and the instantaneous amplitude logarithmic likelihood as it is shown in Figure 23a. 

This low likelihood value is due to an abnormally high (and wide) value of the instantaneous amplitude of the IMF 6 Hilbert Spectrum as shown in Figure 23b, where the dashed gray line indicates the low likelihood time. The IMF 6 sequence also displays high and wide values at the abnormal time, as is depicted in Figure 23c. Finally, Figure 23d shows the evolution of electricity demand (in blue) and weekly demand (in orange), where it can be seen that, at the abnormal demand time, there is a weekly demand in “V”, while one year earlier or later, the weekly demand mainly looks like a “W”.

The abnormal behavior of electricity demand detected analyzing IMF 6 is shown in Figure 24, where a normal week is displayed in panel (a), while three New Year’s weeks are shown in the remaining panels. A normal week (a) usually has two days of lesser demand: Saturday and Sunday. The New Year’s week of 2017 (b) does not very much differ from this pattern. However, 2018 New Year’s week (c) has three days (not just two) of lesser demand. On the other hand, 2019 New Year’s week (d) also has an abnormal pattern, but with 4 days of lesser demand. This longer abnormality is not detected by IMF 6 but, instead, it is caught by IMF 7, probably due to its longer periodicity.

Abnormal demand detection is an important issue in smart metering when an algorithm is incorporated into the meters to detect abnormal electricity consumption as a result of illegal human intervention [12], or when external events produce anomalies in large systems’ energy demand [47,48] or customer-related consumption [49]. 

Some frequency domain-based methods have been proposed for anomaly detection in power load curves either using Fourier Transform [50] or Wavelet Transform [51]. However, to the best of our knowledge, no solution has been proposed using Hilbert–Huang Transform. As has been revealed throughout the paper, HHT-based anomaly detection provides a multiscale perspective, making possible to detect short-, medium-, and long-term anomalies.

A related issue is load curve characterization in which spectral methods have also been proposed using Fourier [27] or Wavelet Transforms [52,53]. No HHT-based solution for load or client profiling has been found in the technical literature. Again, the multiscale perspective provided for Empirical Mode Decomposition deserves to be further explored.

### 4.3. Electricity Demand Sequence Compression

One of the applications of spectral analysis is its capability of compressing the information required to characterize a signal or sequence. Let us consider a sequence x(n) of N values representing, for instance, the hourly electricity demand. Its Fourier Transform (FT) also has N harmonics. In fact, only N/2 harmonics are required as the resulting spectrum is symmetric, but each harmonic is a complex number defined by 2 values, so N values are still required to describe the spectrum. If instead of N values, M coefficients are kept and N−M are discarded, the inverse FT recovers an approximation $\tilde{x}\left(n\right)$. Therefore, by reducing the amount of information representing the sequence, an error is introduced. A common measure of the overall compressing error is the Root Mean Square Error (RMSE), defined as: (26)$$RMSE=\sqrt{\frac{1}{N}\overset{N}{\underset{n=1}{\sum }}{\left[x\left(n\right)-\tilde{x}\left(n\right)\right]}^{2}}.$$

An alternative for the spectral representation of a sequence is the use of EMD. By decomposing the sequence x(n), several IMFs are obtained, each of them with N values. However, recalling the procedure to obtain IMFs described in Section 2.4, only the extrema points are required to fully define an IMF as the remaining points are obtained by a standard cubic spline interpolation. As the number of extrema can be considerably lower than N, a reduction in the number of coefficients required to represent x(n) can be expected.

Applying these ideas to our electricity demand dataset, the results shown in Figure 25 are obtained. In panel (a) the number of coefficients required to fully describe each IMF is displayed. This number is approximately halved in every IMF, as is clearly indicated by the dashed red line in the plot. If a compression is required, only extrema representing lower frequencies IMFs are kept. 

The cumulative number of values required when several IMFs are employed is displayed in Figure 25b. These results, expressed as a percentage of the total number of values in the original sequence (N), clearly reveal that even when using the full set of IMFs, the information required is 34.85% lower than N.

From previous results, it is clear that the electricity demand sequence can be compressed either using a Fourier Decomposition (FD) in its harmonics, or an Empirical Mode Decomposition (EMD) in its IMFs. Comparing the error introduced by both procedures, the results depicted in Figure 26 are obtained. For a very high compression ratio (higher than 90%) both methods offer equivalent performance (FD slightly better). In the medium-range compression ratio, FD obtains smaller errors than EMD. But for zero-error compression EMD clearly outperforms FD.

As a final discussion, let us compare the analysis methods discussed in this section. The first and main goal of any spectral analysis should be identifying a signal’s underlying frequencies (or periodicities). Regarding electricity demand, the frequency discovering capabilities of each method are summarized in Table 1.

It can be seen that the Hilbert–Huang Transform discovers more physically meaningful frequencies with an excellent time-frequency resolution. Additionally, a qualitative assessment of the spectral analysis methods is summarized in Table 2.

## 5. Conclusions

Regarding energy demand frequency-domain characterization, the spectral analysis based on conventional Fourier analysis clearly reveals two harmonics at 365 and at 730 showing the main periodic behavior of the load curve featured by a daily evolution with two humps at noon and mid-afternoon. A third harmonic stands out at the frequency of 52, revealing a weekly periodicity characterized by a higher demand on labor days and lower consumption on weekends. Finally, frequencies at 2 (semester) and 1 (year) appear.

On the other hand, by using Hilbert–Huang Transform, 6 new harmonics have been revealed, centered at the following frequencies: 183 (2-day period); 91 (4-day period); 12 (1-month period); 4.35 (12-week period); 6 (2-month period); and 0.3 (covering the full 40-month dataset). 

From a more qualitative point of view, it can be stated that using the Hilbert–Huang Transform (HHT) for spectral analysis of electricity demand leads to clearer, more intuitive, and more meaningful results than those of more conventional techniques like Fourier or Wavelet transforms. HHT obtains smoother spectra with more defined shapes, revealing physically significant harmonics. Moreover, HHT spectra present an excellent frequency resolution, totally independent of time resolution.

Empirical Mode Decomposition (EMD) required in HHT also fosters a deeper analysis of abnormal electricity demand at different timescales. The set of Intrinsic Mode Functions (IMFs) and their instantaneous amplitude and frequency provides very useful tools to detect consumptions not following normal patterns. 

Applying this analysis, an example of abnormal behavior has been extensively studied (IMF 6) corresponding to an anomalous weekly consumption during 2017 New Year’s week. Further analysis using different IMFs has proven to be useful for discovering other irregularities, for instance: A day with only one energy consumption peak (instead of two) at August 2, 2018 (IMF 3); a 2-week singularity at the beginning of 2018 (IMF 7); and an unusual 2017 second trimester (IMF 9). 

The main advantage of employing EMD-based methods relies on the fact that timescale can be adjusted by selecting a certain IMF. In the effort to obtain the frequency (or period) that best defines every IMF, several techniques have been explored (FFT, HHT, autocorrelation, and probability distributions). A common conclusion arises: IMF main frequency is approximately halved in every step, beginning at 730 times per year (twice a day) for IMF 1.

Additionally, EMD permits information compression, which becomes very significant for lossless sequence representation. A 35% reduction has been obtained for the electricity demand sequence. On the negative side, HHT demands more computer resources (memory and CPU time) than conventional spectral analysis techniques.

In summary, the research presented in the paper has reached three main findings. First, Hilbert–Huang Transform outperforms conventional spectral analysis techniques while providing multiple timescales perspectives. Second, a combination of HHT and Empirical Mode Decomposition has proved to excel abnormal energy consumption detection. Third, EMD provides very good lossless compression ratios.

## Figures and Tables

**Figure 1 sensors-20-02912-f001:**
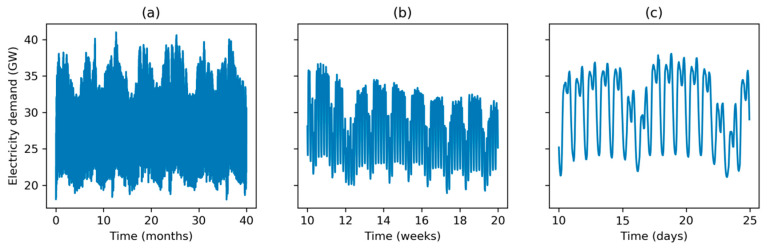
Electricity demand at different timescales. Monthly Hourly Load Values (MHLV) for Spain: (**a**) Overall dataset (40 months); (**b**) 10 weeks’ evolution; (**c**) 15 days’ demand.

**Figure 2 sensors-20-02912-f002:**
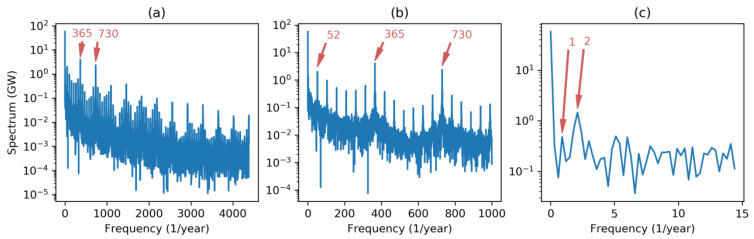
Electricity demand Fourier Spectrum at different frequency scales: (**a**) Full spectrum; (**b**) spectrum discarding higher frequencies; (**c**) low frequency spectrum.

**Figure 3 sensors-20-02912-f003:**
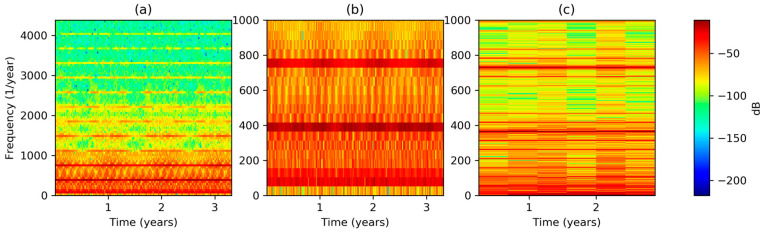
Electricity demand Short-Time Fourier Transform (STFT) spectrogram at different frequency scales and resolutions: (**a**) Full spectrogram, Δt=1 week; (**b**) low frequency spectrogram, Δt=1 week; (**c**) low frequency spectrogram, Δt=0.5 years.

**Figure 4 sensors-20-02912-f004:**
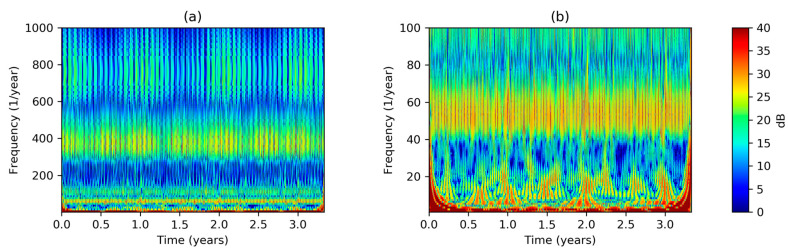
Electricity demand Wavelet Transform (scalogram): (**a**) Including high frequencies; (**b**) low frequency detail.

**Figure 5 sensors-20-02912-f005:**
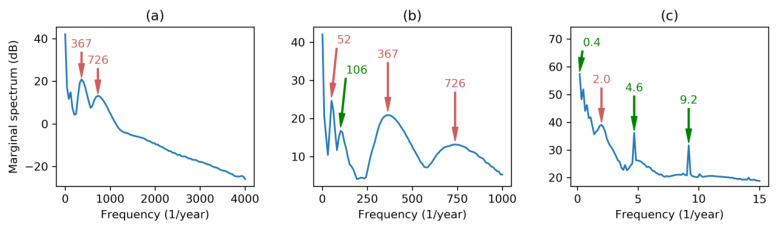
Electricity demand Marginal Wavelet Spectrum: (**a**) Low frequencies; (**b**) medium frequencies; (**c**) high frequencies.

**Figure 6 sensors-20-02912-f006:**
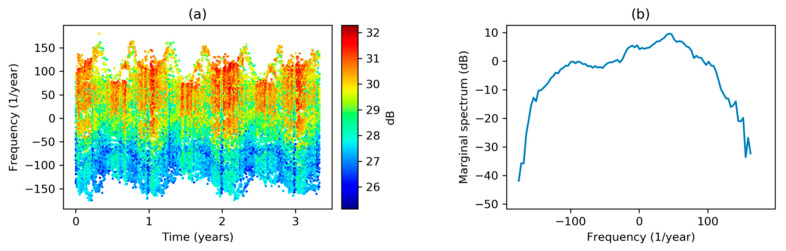
Electricity demand Hilbert Transform: (**a**) Hilbert Spectrum (scatter plot); (**b**) Marginal Hilbert Spectrum.

**Figure 7 sensors-20-02912-f007:**
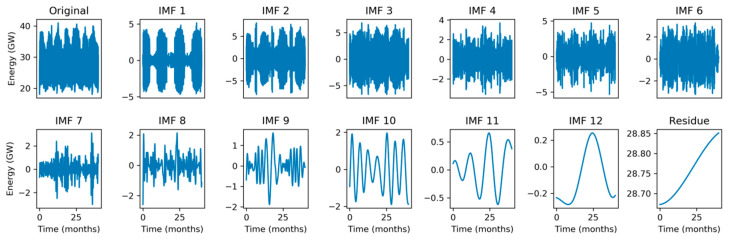
Empirical Mode Decomposition of the electricity demand: Monthly scale.

**Figure 8 sensors-20-02912-f008:**
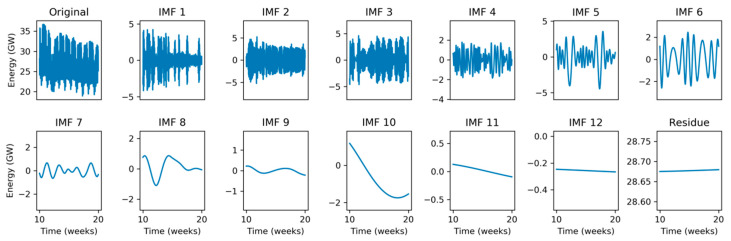
Empirical Mode Decomposition of the electricity demand: Weekly scale.

**Figure 9 sensors-20-02912-f009:**
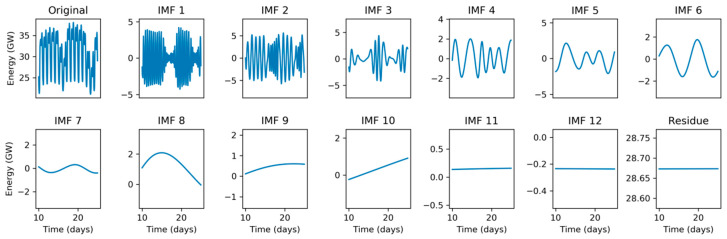
Empirical Mode Decomposition of the electricity demand: Daily scale.

**Figure 10 sensors-20-02912-f010:**
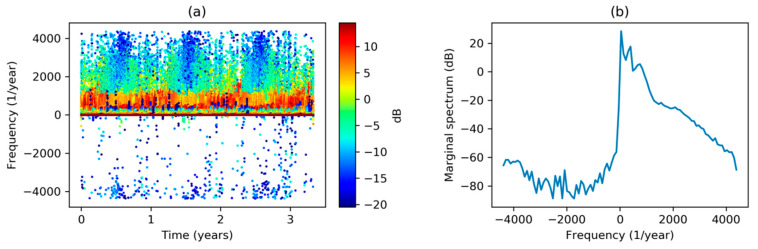
Electricity demand Hilbert–Huang Transform: (**a**) Hilbert–Huang Spectrum (scatter plot); (**b**) Marginal Hilbert–Huang Spectrum.

**Figure 11 sensors-20-02912-f011:**
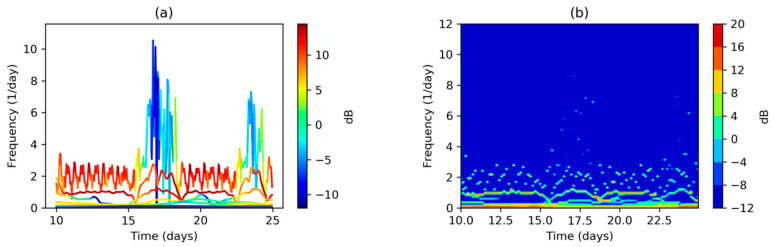
Electricity demand Hilbert–Huang Transform (detail for a 15-day period): (**a**) Hilbert–Huang Spectrum (one line plot for every IMF); (**b**) Discrete Hilbert–Huang Spectrum.

**Figure 12 sensors-20-02912-f012:**
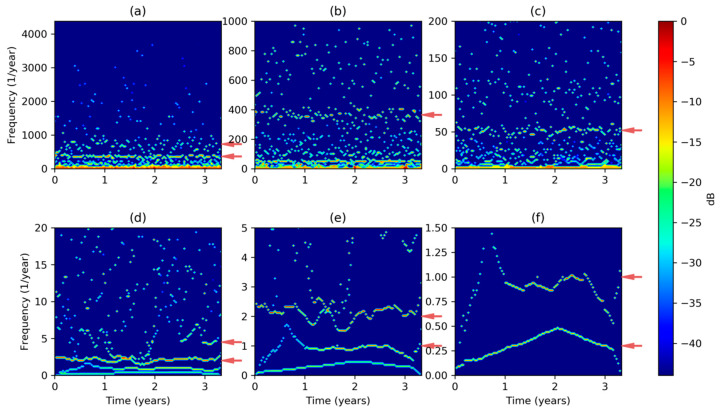
Electricity demand Discrete Hilbert–Huang Transform at different frequency scales: (**a**) Very high; (**b**) high; (**c**) medium high; (**d**) medium low; (**e**) low; (**f**) very low.

**Figure 13 sensors-20-02912-f013:**
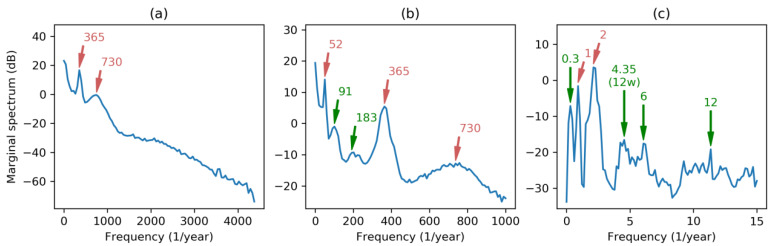
Electricity demand Marginal Hilbert–Huang Spectrum at different frequency scales: (**a**) High; (**b**) medium; (**c**) low. In red are the harmonics obtained by Fourier analysis, while in green are the new harmonics revealed by HHT.

**Figure 14 sensors-20-02912-f014:**
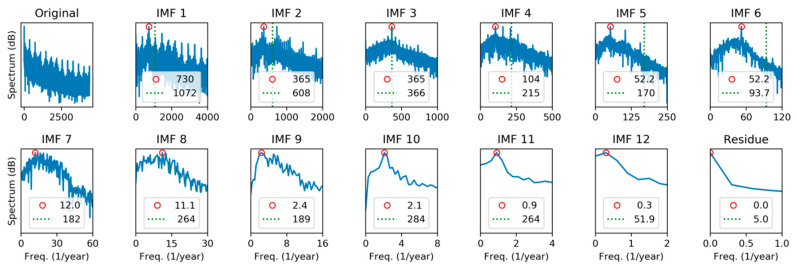
Spectral analysis of the electricity demand Intrinsic Mode Functions (IMFs), using Discrete Fourier Transform. Peak frequencies (red circle) and spectrum centroid (dashed green line) are marked.

**Figure 15 sensors-20-02912-f015:**
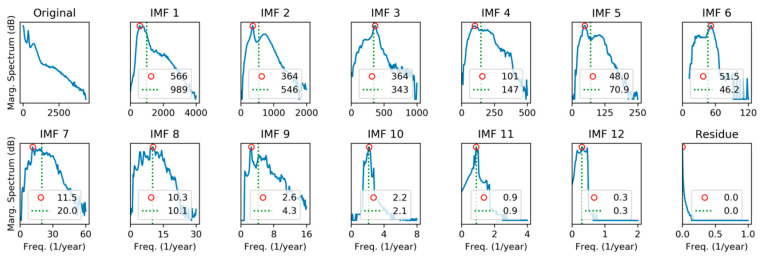
Spectral analysis of the electricity demand IMFs, using Marginal Hilbert Spectrum. Peak frequencies (red circle) and spectrum centroid (dashed green line) are marked.

**Figure 16 sensors-20-02912-f016:**
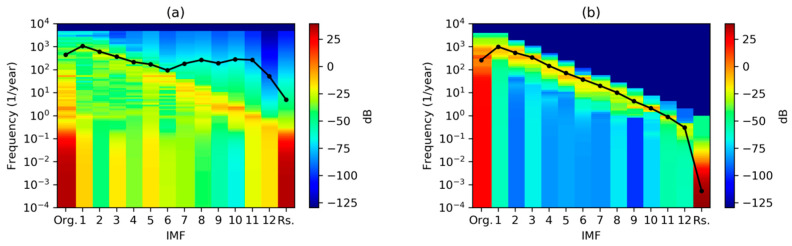
Spectral analysis of the electricity demand IMFs obtained using: (**a**) Discrete Fourier Transform; (**b**) Marginal Hilbert Transform. Black line shows the spectrum centroid.

**Figure 17 sensors-20-02912-f017:**
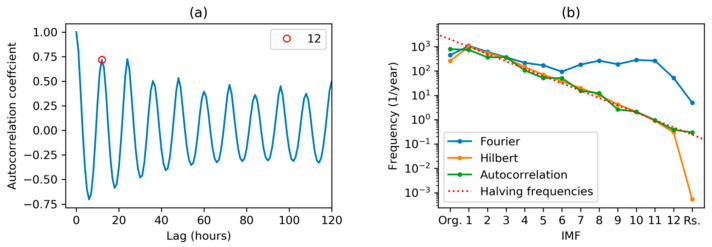
Electricity demand autocorrelation: (**a**) Values corresponding to the first IMF; (**b**) dominant frequencies for each IMF and their comparison to DFT and MHT spectral centroids.

**Figure 18 sensors-20-02912-f018:**
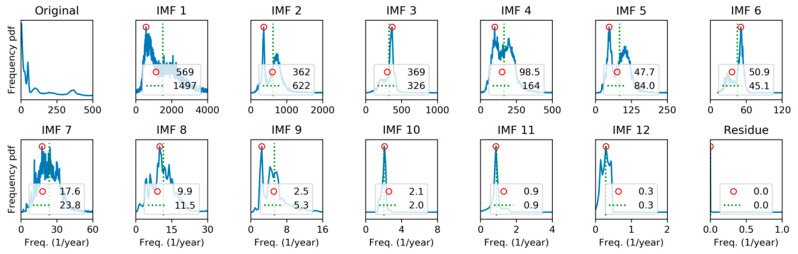
Instantaneous frequency Probability Density Function (PDF) of the electricity demand IMFs. Instantaneous frequency mode (red circle) and mean (dashed green line) are marked.

**Figure 19 sensors-20-02912-f019:**
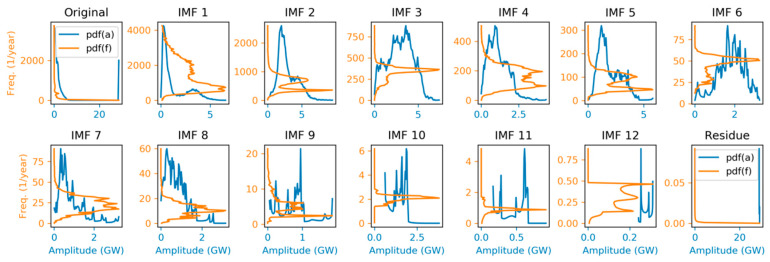
Instantaneous amplitude Probability Density Function (PDF) of the electricity demand IMFs (blue line). In a rotated way (orange line), frequency PDFs (horizontal axis) versus instantaneous frequency (vertical axis) are also depicted.

**Figure 20 sensors-20-02912-f020:**
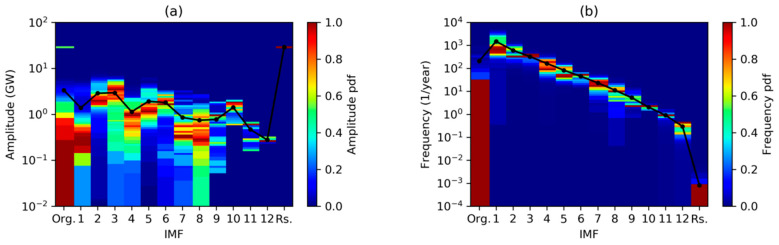
Statistical analysis (probability density function) of the electricity demand IMFs Hilbert Spectra: (**a**) Instantaneous amplitude; (**b**) instantaneous frequency. Black line shows the statistical mean.

**Figure 21 sensors-20-02912-f021:**
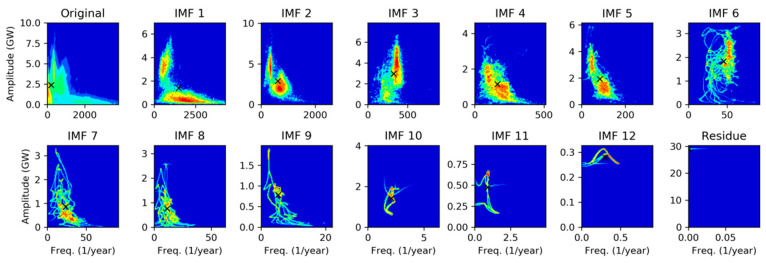
Joint Probability Density Function of the electricity demand IMFs Hilbert Spectra. Joint PDF values in log-scale and normalized in the 0–1 range.

**Figure 22 sensors-20-02912-f022:**
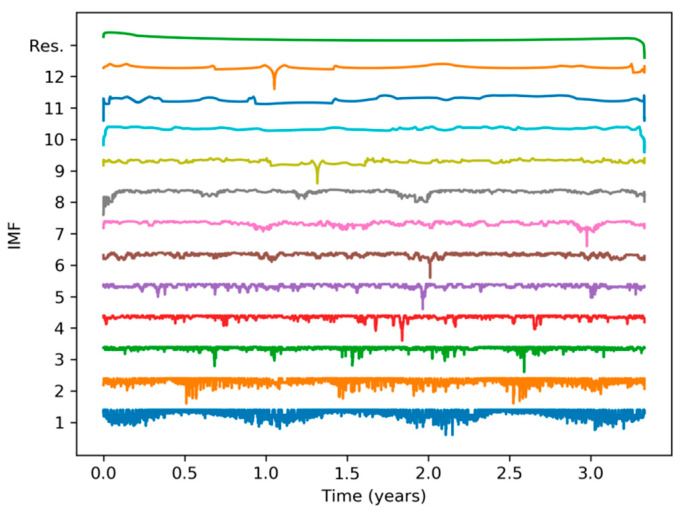
Instantaneous amplitude logarithmic likelihood for every IMF of electricity demand.

**Figure 23 sensors-20-02912-f023:**
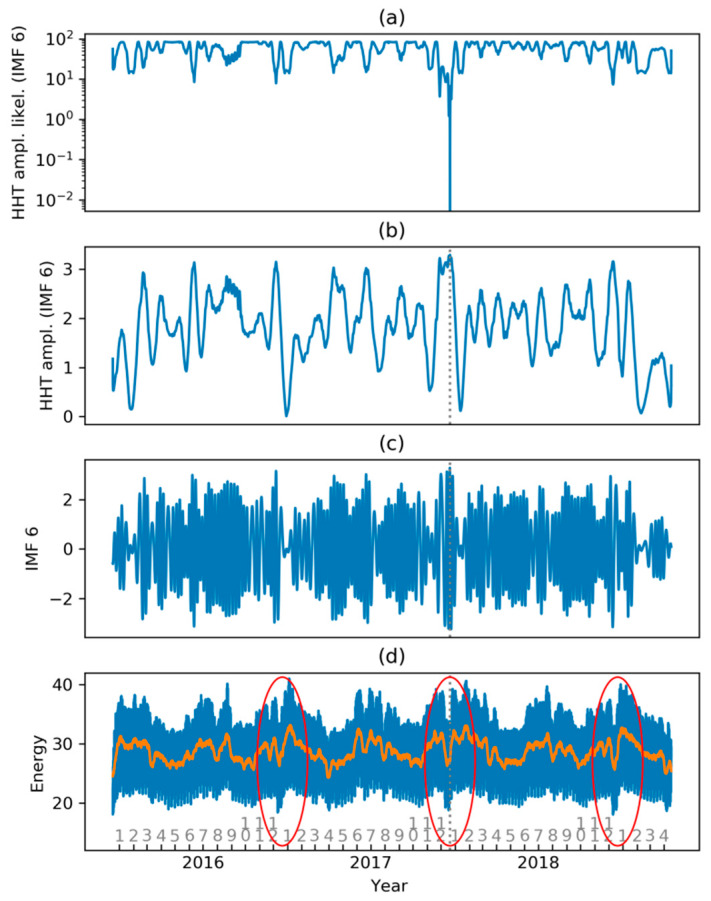
Abnormal behavior of IMF 6: (**a**) Instantaneous amplitude logarithmic likelihood for IMF 6; (**b**) instantaneous amplitude Hilbert Spectrum of IMF 6; (**c**) IMF 6; (**d**) electricity demand (blue) and weekly mean (orange).

**Figure 24 sensors-20-02912-f024:**
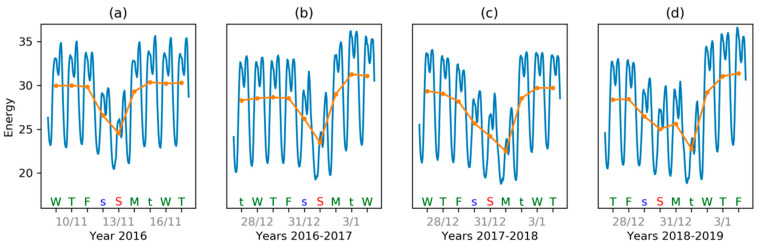
Weekly evolution: (**a**) Normal week; (**b**) 2017 New Year’s week (about normal); (**c**) 2018 New Year’s week (identified as abnormal by IMF 6); (**d**) 2017 New Year’s week (identified as abnormal by IMF 7).

**Figure 25 sensors-20-02912-f025:**
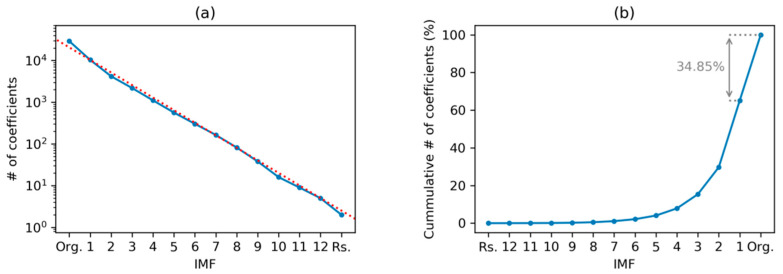
Electricity demand sequence compression: (**a**) Number of coefficients required to fully describe an IMF (dashed red line for halving this number); (**b**) cumulative number of coefficients when several IMFs are employed (as a % of the total number of values in the original sequence).

**Figure 26 sensors-20-02912-f026:**
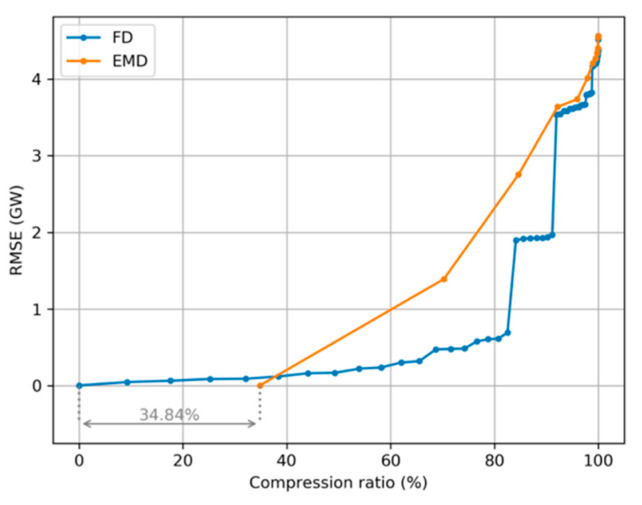
Spectral representation of electricity demand. Root Mean Square Error (RMSE) versus the percentage of discarded values (compression ratio) for Fourier Decomposition (FD) and Empirical Mode Decomposition (EMD).

**Table 1 sensors-20-02912-t001:** Frequencies discovered for different spectral analysis methods.

Periodicity	Theoretical	Fourier	Wavelet	Hilbert-Huang
12 h	730	730	726	730
24 h	365	365	367	365
2 days	182.5			183
4 days	91.25		106	91
1 week	52.14	52.00	52.00	52.00
1 month	12		9.2	12
2 months	6			6
12 weeks	4.35		4.60	4.35
6 months	2	2	2	2
12 months	1	1		1
40 months	0.30		0.40	0.30

**Table 2 sensors-20-02912-t002:** Comparison of spectral analysis techniques.

Feature	Fourier	Wavelet	Hilbert-Huang
Linearity required	Yes	Yes	No
Stationarity required	Yes	No	No
Frequency analysis	Yes	Yes; Marginal WT	Yes; Marginal HHT
Time –frequency analysis	Yes; STFT	Yes	Yes
Time–frequency resolution	Constant	Variable	Arbitrarily small
Spectrum smoothness	Low	Medium	High
Frequencies discovering	Medium	High	Very high
Non-linear modes discovering	No	No	Yes
Compression ratio	Medium	High	Low
Lossless compression ratio	Low	Low	High
Processing requirements	Low	Medium	High

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
