# Peer review of "Spectral Analysis of Electricity Demand Using Hilbert–Huang Transform"

_sensors, 2020, doi:10.3390/s20102912_

Round 1
Reviewer 1 Report
Generally speaking, the idea of the paper is interesting and demonstrates the potential in a scientific journal. Earlier than publication the following comments would be considered:
- The motivation of doing the work is still vague for the reader! Why there is a need for such a work?
- What is the most important finding of the work? In addition to that, there is a need for revising/ extending the conclusion section and stating the results of the study in a better manner.
Author Response
Please find responses attached

Reviewer 2 Report
The article analyzes Fourier Transforms (FT), Wavelet Transform (WT), and Hilbert-Huang Transform (HHT) as methods for deriving information from IoT devices. This is both a relevant and important topic. In addition, the article assesses electrical demands and demonstrates each method's capabilities. The methods are clearly presented and well adapted to the reader. Clearly, this part depends on the reader’s prior knowledge, the authors have reached a good compromise between too detailed information and not enough. Readers unfamiliar with the theory are presented enough background to follow the analyses or to investigate the theory further. The presentation has the reader in focuses and provides necessary information and not too much detailed mathematics.
The result section contains a lot of information contained in several figures. Most of the figures are easy to read. Some figures are small, like Fig 8 and 9. However, the motivation for the presentation is understandable and it is difficult to redesign the figures – which is not recommended. It is recommended to include more text that discusses and explains the figures in the text. This can help the reader understand the figures easier.
Author Response
Please find responses attached

Reviewer 3 Report
This paper investigates the role of HHT approach on spectrum analysis of the electrical demands. The paper is of interest and well organized. However, the authors are recommended to address the following concerns:
-The literature review of the study seems to be incomplete. The authors are encouraged to discuss more studies on the literature.
-In the introduction, the authors address some aspects of the smart grid and renewable energy forecasting. It is recommended to discuss some relevant papers in this area of power system. As an example, the following reference uses autoregressive integrated moving average (ARIMA) as a forecasting approach of electrical demand and renewable energy. You should take a look at these kinds of studies (not limited to the following references) and allocate a paragraph in the review to discuss the role of your approach in such studies in the power system. In this paragraph, you should discuss the role of your suggested approach in smart grid and demand-side management comparing the conventional demand forecasting like ARIMA. I expect to see what is the role of your approach in today’s practical problems of grids with high penetration of renewable power.
[1] Stochastic optimization for retailers with distributed wind generation considering demand response, Journal of Modern Power Systems and Clean Energy
-The authors should spend more time to elaborate on the contributions of their proposed approach. It is recommended to describe the contributions at 2 or 3 bullet points in the last paragraph of the introduction.
-While the study uses some abbreviations, it is a good idea to prepare a nomenclature to describe the abbreviations on the first page of the manuscript. In the current form, it is a bit confusing to find the meaning of abbreviations.
-Although the manuscript discussed different spectrum analysis methods, it failed to conclude the conducted researches. In order to fix this issue, two improvements are recommended. First of all, please make a brief comparison between the methods. It can be in the form of a comparison table. Then, please spend more time in the conclusion section. It cannot convey the main results of the research.
-The results of the research are interesting. But the authors failed to discuss them completely. For example, the study reveals that abnormal demand detection is one of the results of the suggested approach. It is an important issue in smart metering when an algorithm is incorporated into the meters to detect abnormal electricity consumption as a result of illegal human intervention. To me, the suggested approach has important applications in the smart metering of electrical demand. In this way, the authors are highly recommended to investigate and describe the application of their method in real smart power systems. There are other applications in the smart grid that should be pointed out by the authors in the manuscript.
Author Response
Please find responses attached

Round 2
Reviewer 3 Report
The authors have addressed all of my concerns. Therefore, the manuscript can be accepted in the present form.